# Laser Immunotherapy: A Potential Treatment Modality for Keratinocyte Carcinoma

**DOI:** 10.3390/cancers13215405

**Published:** 2021-10-28

**Authors:** Silje Haukali Omland, Emily Cathrine Wenande, Inge Marie Svane, Joshua Tam, Uffe Høgh Olesen, Merete Hædersdal

**Affiliations:** 1Department of Dermatology, Copenhagen University Hospital, Bispebjerg, 2700 Copenhagen, Denmark; emily.cathrine.wenande@regionh.dk (E.C.W.); uffe.hoegh.olesen@regionh.dk (U.H.O.); mhaedersdal@dadlnet.dk (M.H.); 2Wellman Center for Photomedicine, Massachusetts General Hospital, Boston, MA 02114, USA; JTAM3@mgh.harvard.edu; 3Center for Cancer Immune Therapy, Department of Oncology, Copenhagen University Hospital, 2730 Herlev, Denmark; Inge.Marie.Svane@regionh.dk; 4National Center for Cancer Immune Therapy, Department of Oncology, Copenhagen University Hospital, 2730 Herlev, Denmark

**Keywords:** keratinocyte carcinoma, basal cell carcinoma (BCC), squamous cell carcinoma (SCC), laser immunotherapy, laser, immunotherapy, ablative fractionated laser (AFL), immune response, immunohistochemistry

## Abstract

**Simple Summary:**

In light of expanding incidences of keratinocyte carcinoma (KC) with many patients developing multiple tumors, the demand for new treatment modalities is high. With the approval of cemplimab for locally advanced and metastasizing basal cell carcinoma and squamous cell carcinoma, KC is now included as an indication for systemic immunotherapy. At present, however, systemic KC therapy remains limited by the severe side effects associated with treatment. Immunotherapy might be more broadly applied if locally administered. Localized to the skin, KCs are easily accessible to topical drugs and physical interventions such as laser. There is an increasing appreciation of lasers’ potential to activate an immune response. Further enhancement of the laser-based immune activation might be obtained by combining laser and immunotherapeutic agents, known as laser immunotherapy. In search of new treatment modalities for KC and aiming to broaden the field of KC immunotherapy, this review discusses the current literature on immune activation following both laser monotherapy and laser immunotherapy.

**Abstract:**

The role of the immune system in cancer growth is well recognized and the development of immunotherapy represents a breakthrough in cancer treatment. Recently, the use of systemic immunotherapy was extended to keratinocyte carcinoma (KC), specifically locally advanced and metastasizing basal and squamous cell carcinoma. However, since most KC lesions are non-aggressive, systemic treatment with associated side effects is rarely justified. Conversely, topical immunotherapy with imiquimod remains restricted to premalignant and superficial lesions. Use of laser in the treatment of KC has evolved from physical tumor destruction and laser-assisted drug delivery to laser-mediated immune modulation. Evidence indicates that laser monotherapy can lead to immune cell infiltration, tumor reduction and resistance to tumor re-inoculation. Combining laser with immunotherapeutic agents, termed laser immunotherapy (LIT), may further potentiate immune activation and tumor response. Studies on LIT show not only direct anti-tumor effects but systemic adaptive immunity, illustrated by the prevention of tumor recurrence and regression in distant untreated tumors. These findings imply a therapeutic potential for both local and metastatic disease. This work provides rationales for immune-based treatment of KC and presents the current status of KC immunotherapy. Aiming to expand the field of KC immunotherapy, the review discusses the literature on immune activation following laser monotherapy and LIT.

## 1. Introduction

In oncology, the role of the immune system in cancer prevention and control is well-recognized, and the introduction of systemic immunotherapeutics has revolutionized clinical cancer treatment. Keratinocyte carcinomas (KCs), however, differ from many cancer types in that most tumors remain localized, with low metastasizing potential. Thus, only few patients with aggressive disease are candidates for systemic immunotherapy, and associated treatment toxicity remains a major limiting factor. If locally administered, immunotherapy might be more broadly applied to treating KCs. The cutaneous localization of KC renders this cancer type easily accessible to topically applied drugs, as well as physical interventions such as laser.

Dermatologists were among the first medical specialists to incorporate lasers in medicine, where treatment of skin cancer was an early indication of interest [1]. For decades, the focus of laser-based treatment of KC, comprising basal cell carcinoma (BCC) and squamous cell carcinoma (SCC), has been on the modality’s tumor destructive effects and closure of vessel supply [2,3]. Clinical application of laser therapy for KC has since broadened with introduction of fractional laser-assisted drug delivery, a technique which enhances topical delivery of drugs through the upper skin layers [4,5]. Now, beyond causing physical tumor destruction and facilitation of cutaneous drug distribution, there is an increasing appreciation of lasers’ potential to activate an anti-tumoral immune response through controlled tissue injury. Ideally suited to treat tumors freely accessible on the skin, lasers’ impact on local immune environments might be harnessed to treat KC as illustrated in Figure 1 where a BCC is treated with an ablative fractionated laser (AFL) (Figure 1).

Termed laser immunotherapy (LIT), the concept of combining immune and laser therapy, has multiple potential advantages, including enabling topical delivery of immunological agents, as well as laser-based amplification of immunotherapeutic agents. This work presents rationales for use of immune-based treatment of KC and examines the current status of KC immunotherapy. While the term KC includes both BCC and SCC, it is important to state that these tumors differ both in terms of clinical presentation and aggressiveness as well as in their biological evolution. Impairment of the sonic hedgehog pathway plays a key role in BCC pathogensis, the most prevalent of the KCs. BCC display very low metastazising potential, while SCC metastazises in 4–6% of cases [6]. While surgical excision and radiofrequency are accepted treatments for both BCC and SCC, topical immunotherapy is restricted to BCC. Systemic immunotherapy, however, is now approved for both tumors where conventional treatment is inadequate due to severe disease. Aiming to expand the field of KC immunotherapy, the review discusses the current literature on immune activation following both laser monotherapy and LIT. Studies included in the review comprise direct assessments of LIT in patient KC and KC models as well as studies in other tumor models and healthy/photodamaged skin.

## 2. Rationales for Immunotherapy in Keratinocyte Carcinoma

The immune system’s ability to recognize and eliminate transformed malignant cells is well established. Improved understanding of tumor pathophysiology and the role of the immune system in tumor control, has led to the development of systemic immunotherapy; one of the most important breakthroughs in modern medicine for treatment of various aggressive cancers, including malignant melanoma (MM) [7].

In the context of KC treatment, two biomolecular rationales support the use of immune check point inhibitors: (1) the presence of programmed death-1 (PD1) on T-cells or programmed death ligand-1 (PD-L1) on cancer cells and suppressive immune cells in tumor tissue and (2) the high mutational burden of KC. Currently, these markers are considered to be among the most valid general predictors of response of immune check point inhibition.

Two larger immunohistochemical (IHC) studies focusing on PD1/PD-L1 in BCC showed positive staining in the majority of tumor cells and tumor infiltrating lymphocytes (TILs), indicating a potential for response [8,9]. It should be noted however that a phase 2 study on cemiplimab (anti-PD1) in SCC showed clinical responses irrespective of baseline PD-L1 status [10]. More specific focus on a subgroup of PD1 positive TILs, namely regulatory T-cells (T-regs), is now appreciated to be a predictor of treatment response. Accordingly, melanoma patients who demonstrated a rapid decline in circulating PD1-positive T-regs upon anti-PD1 treatment were at reduced risk for disease progression [11]. A study on BCC tumor environment has revealed increased T-reg/CD3 ratio in the tumor microenvironment, a feature that is suggested to play a role in tumor escape and further supports the concept of immune checkpoint inhibition in KC management [12]. Whether these T-regs are PD1 positive remains to be elucidated but might prove important, since PD1 signaling is involved in T-reg homeostasis. Interestingly, a previous preclinical study has shown PD1–deficient T-regs to possess increased immunosuppressive activity compared with PD1–intact T-regs [13], indicating that lack of PD-1 signaling enhances the immunosuppressive function of T-regs. Likewise, murine PD-1 deficient T-regs have been shown to be more proliferative and immunosuppressive compared with PD1 intact T-regs [14].

The second emerging biomarker predicting the outcome of checkpoint inhibitors is the tumor mutational burden (TMB) [15]. TMB is a quantitative measure of the number of gene mutations inside cancer cells and is an indirect measure of tumor-derived neoantigens. It is hypothesized that the higher the number of neoantigens within a tumor, the higher probability of target of recognition exists within the tumor for anti-tumor immune response. Genome studies have revealed that KCs have the highest mutational burden of all human cancers, providing another argument for KC immunotherapy [16]. A case series including eight patients with metastatic BCC, four of whom received anti-PD1, presenting the genomic correlates on advanced/metastatic BCC treated with anti-PD1 revealed biological features (high TMB; PD1/PD-L1 amplification) predictive of immunotherapy benefit [17].

The role of the immune system in KC development and maintenance is underscored by substantially higher BCC and SCC rates in immunosuppressed versus immunocompetent individuals [18,19].

## 3. Keratinocyte Carcinoma Immunotherapy: Current Status

The clinical development of immune checkpoint inhibitors has drastically expanded within the last decade, both in terms of new drugs and perhaps more markedly, cancer indications [7,20]. Cemiplimab, a PD1 inhibitor, is the first immune check point inhibitor approved for the treatment of KC of the skin. The drug is authorized for the treatment of locally advanced and metastasizing SCC. In these tumors, cemiplimab demonstrates durable, clinically significant efficacy with an objective response in 44% [10] and 47%, respectively, [21] and an acceptable safety profile. Most recently, cemiplimab was approved for locally advanced and metastatic BCCs either previously treated with a hedgehog pathway inhibitor, or in patients where hedgehog pathway inhibitor is inappropriate. The overall response rate of cemiplimab appears lower for BCCs than SCCs reported in one study as 21% (6/28) in metastatic BCC patients, with no complete responses. In patients with locally advanced BCC, the objective response rate is 29% (24/84), with 6% (5/84) complete responders (trial ID: R2810-ONC-1620).

In addition to cemiplimab studies, evidence of a clinically relevant potential for anti-PD1 treatment against KC has been reported in patients with MM on anti-PD1 treatment. In that population, lower incidences of BCC compared with patients with MM not receiving anti-PD1 was shown. No difference in SCC-incidence, however, was found. Given the aforementioned differences in response rates to cemiplimab for BCC and SCC, the lack of impact regarding SCC incidence is surprising. This could reflect overall lower incidences of SCC compared with BCC, resulting in small sample size [22]. Additionally, patients with metastasizing BCC have been found to show partial or near-complete response to anti-PD1 in five case reports [23,24,25,26,27]. Most recently, a study on the effect of PD-L1-directed vaccination in 10 patients with BCC was published. Vaccinations resulted in vaccine-specific immune responses detectable in blood samples from nine of 10 patients and in skin samples from five of nine patients, suggesting that a PD-L1 vaccine might be effective against some BCCs with minor adverse reactions [28].

Systemic immunotherapy comes with a significant risk of side effects, the seriousness of which must be outweighed by cancer aggressiveness. Since most KCs are localized skin tumors often arising in elderly patients with comorbidities, systemic immunotherapy is reserved for a minority with locally advanced or metastasized disease. In comparison, topical therapy is usually associated with a more tolerable side effect profile. Imiquimod is an approved topical immunotherapy for KC that is associated with markedly fewer systemic side effects. The agent’s use is however restricted to superficial BCC [29] and actinic keratoses. Imiquimod is a toll-like receptor (TLR) agonist that binds to TLR 7 and 8 present on innate immune cells to produce anti-viral and anti-tumoral effects. The drug stimulates plasmacytoid dendritic cells to release INF-α [30] and leads to influx of CD8 positive T-cells, B cells as well as macrophages [31]. Seeking to broaden the treatment indication of imiquimod, combination treatment with physical tumor treatment has been introduced; imiquimod combined with cryotherapy showed promising efficacy for BCC and in situ SCC, with combination therapy being more effective than either treatment alone [32,33]. Going forward, combined imiquimod with laser may exploit not only laser’s destructive effects, but also the modality’s potential for immune activation, conceivably leading to enhanced immunotherapeutic effects.

## 4. Immunological Responses to Laser: Preclinical Evidence

The following section reviews existing literature on laser-induced immune responses. Studies on laser monotherapy that perform objective assessment of immune responses including evaluation of immune cells based on immunohistochemistry and/or flow cytometry and/or qRT-PCR were examined. The review identified reports on both fractional ablative lasers including the CO_2_ laser operating 10,600 nm and Erbium:Yag (Er:YAG) laser at 2940 nm, as well as the non-ablative Nd:Yag laser. A total of 11 articles were identified, of which one was in a UV-induced SCC-mouse model. Of these, six represented preclinical animal studies (mice, rats and pigs), one a preclinical human in vitro skin model and four clinical patient trials. In five of six preclinical studies, fractional CO_2_-laser was applied, while one study used an Nd:Yag laser. In all the preclinical CO_2-_studies density was 5%, underscoring that tumor was not physically removed by the laser. In the four clinical investigations, three studies applied CO_2_-laser and one study included Er:Yag laser just as Er:Yag laser was applied in the in vitro skin model study (summarized in Table 1).

Overall, studies on laser-induced immune response showed a cascade of locally infiltrating immune cells at different time points after laser exposure (Figure 2). Discussed in detail in the following section a tendency towards early influx of neutrophilic granulocytes followed by an extended period of lymphocyte and macrophage attraction, irrespective of study design and applied laser device, was shown.

Within 1–48 h of AFL-therapy, preclinical animal studies suggest induction of notable innate immune responses. Among the innate immune actors, neutrophilic granulocytes have been shown to dominate infiltrating immune cell populations in response to laser-mediated thermal injury [34,35,36]. Shown in healthy porcine skin and murine SCC tumors, granulocyte infiltrates are identified as early as 5 h following AFL and can remain present as long as 14 days post-irradiation, respectively, with indication of a primarily N1 antitumorigenic neutrophil profile [34]. Proteins involved in immune cell communication such as interleukin-6 (IL6) and monocyte chemoattractant protein-1 [37] (MCP-1) have likewise been implicated in the early stages following AFL, although their role appears more multifaceted. IL-6 as a key player in the activation, proliferation and survival of lymphocytes during active immune responses [45] and is involved in the regulation of acute phase-responses [46]. Meanwhile, MCP-1 leads to monocyte chemotaxis and T-lymphocyte differentiation via binding to the CC chemokine receptor 2 [46]. Both IL6 and MCP-1, however, are proteins considered tumor promoting in the tumor setting [47], which stands in contrast to their role as acute immune regulators. Their significance in the skin environment immediately following AFL exposure remains unknown.

Early increases of innate immune cells evolve into a more adaptive signature in the later stages of AFL intervention. After 48 h, increases in lymphocytes, altered T-reg vs CD8 ratios, increases in macrophages and growth factors are reported in a range of studies. Shown in mice, increased CD3, CD4 and CD8^+^ T-cell lymphocyte infiltration occurred in intradermally injected tumors of the thigh established by inoculation of a wild-type colon carcinoma cell line following AFL exposure [38]. Interestingly, AFL-induced tumor-specific CD8^+^ T-cell responses have been shown in two experiments [34,38]. The potential for laser-induced tumor-specific lymphocyte infiltration in at least some instances thus appears possible.

Some studies point to an increased ratio of CD8 versus T-reg following AFL [34,38]. T-regs are recruited as a subpopulation of TILs and are generally considered suppressors of tumor-specific antigen-response. In one study, AFL treatment abrogated the increase of immunosuppressive T-regs and led to expansion of tumor-specific CD8 positive T lymphocytes. The suppressive activity of T-regs was confirmed by a cytotoxicity assay, showing the T-regs suppress the cytotoxic activity of the antigen-specific CD8+ T cells from AFL treated mice [34]. This indicates the potential of laser to abolish inhibitory anti-tumor signaling of the tumor microenvironment.

Laser-induced ablation and thermal injury is shown to cause infiltration of macrophages. This is expected since macrophages are important modulators of wound healing where they are involved in the transition from inflammation to proliferation phase [48]. Investigations of AFL demonstrate increased and sustained infiltration of macrophages in pig skin up to 120 h [35] after laser as well as in scar tissue up to 96 h following AFL treatment [37]. This laser-induced increase in macrophage infiltration is thus likely indicative of a wound healing process.

Changes in transforming growth factor β (TGF-β) [37], a cytokine playing an important but complex role in immunoregulation, has also been linked to laser treatment. In a study investigating inflammatory response and matrix remodeling in scar tissue following AFL intervention, enhanced TGF-β was shown up to 168 h postintervention. Like IL-6 and MCP-1, TGF-β exerts both tumor promoting as well as tumor suppressing roles. In the tumor microenvironment, TGF-β inhibits host tumor immune surveillance [49] while TGF-β is involved in wound healing and matrix remodeling in wounds. Thus, the significance of the cytokine in relation to laser therapy remains unclear.

In addition to AFL-induced infiltration of immune cells, concurrent assessment of tumor reduction, clearance and resistance to tumor re-inoculation is reported in three murine AFL studies. One study demonstrated that neutrophil and lymphocyte infiltration in mice inoculated with a colon carcinoma cell line coincided with complete tumor remission in almost half of AFL-treated mice. Furthermore, rejection upon re-inoculation of the same tumor cell line in AFL-cured mice was achieved, indicating adaptive immunity with prevention of tumor recurrence [34]. Importantly, antitumor effects have also been shown for KC tumors. Thus, one murine study demonstrated that AFL exposure not only induced lymphocyte infiltration but led to size reduction in 75% of treated SCCs, while 58% showed complete tumor clearance [36]. There is some evidence to suggest that the mechanism of tumor rejection and recurrence prevention is specific. One study showed that blockade of CD8 lymphocytes negated AFL-induced increased survival. Furthermore, depletion of CD8 cells prior to AFL led to failed tumor reduction [38]. The role of CD8 cells following AFL intervention in terms of antitumor response thus represents an intriguing avenue of further cancer research.

Head-to-head comparisons of laser versus surgical excision provide an interesting insight into potential immune activating effects also of non-ablative laser devices, including the Nd:Yag 1064 nm laser. In a study of rats bearing two inoculated liver tumors in which one was treated with either laser or partial hepatectomy, and the other left untreated, non-ablative laser, but not surgery, led to tumor shrinkage of the second untreated tumor [39]. Unlike rats allocated to surgery, the laser group demonstrated expression of CD8 and B7-2 (CD86) at the untreated tumor border, as well as reduced peritoneal spread [39]. These findings suggest that lasers may facilitate an antitumor response that might be equivalent to vaccination since laser-treated tumor cells in opposite to surgically excised tumor cells are left in situ. Laser-facilitated, exposed tumor antigens might stimulate antitumoral immune response.

## 5. Immunological Responses to Laser: Clinical Evidence

The clinical studies reporting on laser-induced immune activity vary in study design and investigated immune markers. In contrast to the preclinical studies where most studies report on cellular infiltration, clinical studies primarily describe laser-mediated alterations in chemokines [44], interleukins [44], lymphocytes [40,42], growth factors [40,41], TLRs [43] and heat shock proteins (HSPs) [40].

In clinical studies where CO_2_-laser was examined increases in TGF-β are shown, as seen in the previously discussed preclinical matrix remodeling study [40,41]. Early increases of TGF-β are reported within 24 h of laser exposure, peaking at day 3 [40,41]. Similarly at day 3, increased mast cell and lymphocyte recruitment are identified [42], including an influx of CD3 and CD20 positive lymphocytes [40]. All studies reporting on increased TGF-β were conducted in wound healing/scar tissue settings, emphasizing the importance of this growth factor to tissue healing. In addition to TGF- β, high expression of HSPs following AFL with sustained expression 14 days following intervention is demonstrated in one study [40]. HSPs are soluble intracellular molecules constitutively expressed under physiological conditions and increased expression in response to physical stimuli such as heat. HSPs are involved in innate immunity by elicitation of nonspecific cytokine and chemokine secretion as well as in acquired immunity with the provision of peptides for MHC-loading and antigen specific responses [50].

The Er:Yag laser has also been investigated in the clinical setting. Results are varied, favoring both inhibited and enhanced immune reactions. In one study assessing the immunology of healthy skin after laser treatment, reduction of Langerhans cells and TLR2 and TLR9 but increased TLR3 expression on day 7 was shown [43]. TLRs are expressed in innate immune cells such as dendritic cells and macrophages and play crucial roles in the innate immune system with bridging to adaptive immunity. TLR3 stimulation is suggested to skew M2 macrophages (pro-tumorigenic) to the anti-tumorigenic M1 phenotype [51]. This was supported by administration of TLR3 agonist in a murine tumor, leading to shift in macrophage phenotype from M2 to M1 and regression of tumor growth [51]. A second study on Er:Yag laser in an in vitro human skin model resulted in upregulated expression of the chemokines CXCL1, CXCL2 and CXCL5 and increased expression of the interleukins IL6, IL8 and IL24 [44]. These three chemokines are known to attract and activate neutrophilic granulocytes during acute granulocytic inflammation. Similarly, IL8 is part of the CXC family involved in attraction and activation of neutrophils. If secreted by tumor cells, it has been associated with advanced tumor stage. Furthermore, IL24 belongs to the IL10 gene family that possess antitumor properties including inhibition of tumor angiogenesis and induction of tumor cell apoptosis [52]. In the same study, interleukins IL18 and IL36β both belonging to the IL1 family, were on the other hand downregulated [44], again emphasizing immunoreactions not necessarily pointing in one direction. IL18 is involved in the production of interferon-ɣ acting as a costimulatory factor to IL12. High IL18 concentrations has been associated with advanced tumor stages in a variety of cancers [53].

The immune activating feature of lasers is furthermore supported by their potential to act as vaccine adjuvants, recently reviewed for ultrashort pulsed laser, non-pulsed laser, AFL and non-ablative fractional lasers [54]. The review covers a body of more than 25 publications covering preclinical and clinical studies showing substantial laser-mediated induction of specific antibodies towards vaccine antigens or infiltration of cytotoxic T lymphocytes in tumor [55] and infection models [56] with many studies pointing at a synergistic effect of laser and chemical adjuvants. Both common and specific mechanisms across different laser types were observed with laser increased motility of antigen presenting cells being one of the suggested mechanisms of augmented immunity [57]. The review concludes that laser as a vaccine adjuvant can safely augment not only humoral but also cell-mediated immune responses, with certain advantages over traditional vaccine adjuvants.

In summary, evidence indicating immediate attraction of neutrophils followed by later influx of lymphocytes and growth factors suggests that lasers can substantially impact on both innate and adaptive immune activity. In preclinical studies, this pattern of early neutrophilic leukocyte increases followed by a more protracted attraction of lymphocytes was distinct, indicating in some instance specific anti-tumor responses. More conflicting immune reactions including both inhibited and enhanced immune reactions were noted in the clinical studies, all of which, were conducted in healthy/sun-damaged skin. In this context, the complex role of interleukins cannot be understated. Cytokines are molecular messengers enabling immune cells to communicate playing key roles in regulation of the immune system. In the reviewed studies where most investigations were performed on non-tumor tissue, altered cytokine secretion following AFL should be considered a sign of innate immune activation rather than mediators of a tumor-specific response in the tumor microenvironment. Although the understanding of the immune modulating impact of laser treatment in oncology is relatively new, current preclinical results seem encouraging so far.

## 6. Laser Immunotherapy

As discussed in the previous section, laser treatment leads to substantial activity of the immune system. Provided as monotherapy for cancer, however, lasers are commonly inadequate. It is conceivable that enhanced efficacy could be obtained by combining laser with immunostimulatory agent or drug as illustrated in Figure 3 providing a conceptual summary of laser monotherapy and LIT.

In the following section, primarily preclinical studies reporting on combination treatment with objective assessment of immune response are reviewed. Included are studies which performed either a clinical evaluation or an immune cell measurement based on immunohistochemistry and/or flow cytometry and/or qRT-PCR. A total of 13 articles on LIT were reviewed, with 11 being preclinical animal studies (mice, rats) and three being human case reports/series (one study including both mice and one patient with SCC). In nine of the 13 studies, near-infrared (NIR) laser was investigated, while three studies reported on CO_2_-laser, and one included Nd:Yag laser. All included studies were conducted in animals or patients with tumor bearing tissue with two studies being in SCC tumor models. Applied immunostimulatory agents consisted of intratumorally injected chitosan gel, while later studies applied topical imiquimod, systemic anti-PD1 and/or anti-cytotoxic T-lymphocyte antigen-4 (CTLA4) (Table 2).

First proposed in 1997, the concept of LIT combines the immune activating effects of lasers and immunostimulatory agents. Initially, LIT employed NIR lasers, but has expanded in recent years to include ablative lasers (including CO_2_ and Er:Yag lasers) and the non-ablative Nd:Yag laser. Pioneering LIT studies employed NIR laser, generally reporting reductions in tumor burden, increased survival and tumor immunity. For example, NIR-based LIT in combination with either the immunostimulant agent chitosan gel in rat mammary tumor [58,59,60,61] or imiquimod in UV-induced SCC in mice [62], resulted in infiltration of lymphocytes and plasma cells [61], decreased tumor burden and increased survival [58,60,62] with combination therapy being superior to either treatment alone [60,62]. Interestingly, resistance to tumor rechallenge [58] and shrinkage of non-treated tumors [62] was shown. An abscopal effect demonstrated by tumor growth resistance, was further seen in naïve rats with adoptive transfer of splenocytes from LIT-cured rats [59].

On its own, chitosan gel has been shown to cause immune activation via increased secretion of TNF-α by mouse macrophages [69]. Chitosan has furthermore been proved effective as a vaccine adjuvant [70]. No direct comparison of chitosan gel with other immunostimulatory drugs have been performed. One could however speculate that the immune activating effects of chitosan are modest compared to approved, immune activating drugs.

AFLs are increasingly favored in LIT literature. Three recent AFL studies [36,38,63] show promising effects, including both direct tumor inhibition, as well as indirect systemic immunity. One murine study of tumors established by inoculation of wild type colon carcinoma cells, reported use of fractional CO_2_ laser in combination with systemic anti-PD1 [38]. Both laser monotherapy as well as combination therapy led to significantly slower tumor growth compared to untreated control. Indicating establishment of systemic immunity, shrinkage of untreated contralateral tumors in the same animal occurred in the combination therapy group only. Equivalent to NIR studies, AFL-based LIT led to long-term immunity since re-inoculated tumors failed to grow in previously treated mice. This was the case for all mice treated with combination therapy (AFL and anti-PD1) and all but one mouse treated with AFL-monotherapy. Tumors in the naïve mice all progressed [38].

Similar results have most recently been published for poorly immunogenic wild-type melanomas. As previously discussed, the effect of immune checkpoint inhibition therapy correlates to TMB. In tumors with low TMB, an attempt to enhance efficacy of immune checkpoint inhibitors was sought using LIT with AFL and imiquimod [63]. Bilateral flank tumors were initially established in mice via inoculation of a low neoantigen burden melanoma line. Mice were subsequently treated with combinations of AFL, imiquimod and immune checkpoint inhibitors. Imiquimod and AFL was found to improve systemic checkpoint blockade, with rates of complete tumor regression increasing from 0% with any monotherapy, to 10% with combination of any two treatments, to 50% upon triple therapy with AFL + imiquimod + anti- PD1. Furthermore, a significant increase in intratumoral CD3^+^ T-cell infiltration was shown after triple therapy. Importantly, almost identical responses were seen in both tumors despite only applying laser and imiquimod unilaterally, highlighting an abscopal effect of combination treatment [63].

In contrast to many other cancer types including a subgroup of melanomas, UV-induced KC are characterized by a high number of neoantigens. In a recently published study on UV-induced murine SCCs, LIT consisting of AFL and imiquimod was found superior to either treatment alone, demonstrated by tumor shrinkage and 75% tumor clearance at day 14. Furthermore, combinatory treatment led to the earliest and most potent local innate as well as the most robust adaptive skin cell infiltration [36]. This is of high clinical relevance for the treatment of KC, since this is the first study of its kind in a UV-induced SCC-model.

In most of the referred studies, LIT has been performed at a single time point. However, in clinical cancer settings, systemic immunotherapy is given at repeated intervals, typically every 2–3 weeks depending on tumor type, patient performance status, side effects and tumor response [71]. Aiming to mimic clinical treatment settings, three murine studies applied nanoparticles as part of a LIT strategy, obtaining tumor shrinkage, tumor immunity as well as increased survival. Specifically, drug encapsulation of anti-PD1 in 40-day, sustained release nanoparticles, resulted in almost complete tumor elimination of primary tumors as well as effective growth inhibition of distant untreated murine tumors [64]. Similarly, LIT performed using a combination of nanoparticle encapsulated TLR agonist and intravenous cytotoxic T-lymphocyte antigen (CTLA-4), led to systemic effects including near-complete inhibition of tumor growth in non-treated tumors in mice with bilateral tumors. Moreover, strong immune-memory effects mimicking cancer vaccination was observed 40 days after treatment, indicating the possibility of preventing tumor relapse with the method [65].

Not only encapsulation of immunostimulatory drugs, but also of exogenous light-absorbing agents for the purpose of highly selective tumor ablation, has been investigated. A laser light-absorbing nanoparticle was combined with intraperitoneally injected anti-PD1 in a melanoma mouse model, resulting in delayed tumor growth and increased infiltration of antigen-presenting cells and CD8+ T cells. Furthermore, significantly increased survival was observed in the LIT treated group compared to individual treatments alone [66].

As evidenced in this review, most LIT research has been performed in the experimental animal setting. At present, clinical use of LIT remains limited. In the existing literature, case studies of patients with incurable MM treated with LIT, consisting of imiquimod and NIR laser, suggest clinical benefit with local tumor response [67,68]. Combined imiquimod and NIR has also led to tumor shrinkage and increased inflammatory infiltrate in a patient with refractory SCC [62].

Apart from acting as immune stimulator, AFL also enables cutaneous application of a large variety of drugs otherwise not permeable to the skin. Laser assisted drug delivery is a well-documented physical enhancement technique that enables effective cutaneous uptake of various substances [72]. Indeed, strong evidence already supports the use of laser assisted drug delivery for SCC precursors such as actinic keratosis and Bowens disease [73]. In this paper’s context, it is worth noting that successful in vitro, topical delivery of the PD1 inhibitor nivolumab was recently published [74], providing further support of the feasibility of topical AFL-based LIT in the treatment of KC.

## 7. Conclusions

In oncology, combination therapy represents a conceptual cornerstone to reduce drug resistance and boost therapeutic effectiveness. LIT, combining lasers with an immunostimulant, has the potential to enhance the established immune activation of lasers alone. Various studies have investigated LIT primarily in experimental animal settings with favorable results. Several immunostimulatory agents have been applied in LIT, ranging from the systemic immune checkpoint inhibitors anti-PD1 and anti-CTLA4, to topical TLR agonists such as imiquimod as well as the immunostimulant chitosan gel. Preliminary results indicate not only responses in directly treated tumors, but also systemic effects in distant lesions. These observations indicate a therapeutic potential for metastatic disease. Importantly, a rationale for LIT over laser monotherapy is the induction of immunological memory, mimicking e.g., cancer vaccination.

At present, there is a growing appreciation for the role of tumor microenvironment in cancer therapy and prognosis. In contrast to surgical removal of peritumoral tissue, laser-treated peritumoral tissue is left in situ. In the context of KC, this feature may prove important, since the KC peritumoral skin is often sun damaged and potentially field cancerized. Although speculative, one could surmise that LIT intervention on peritumoral sun damaged skin might prevent further skin cancer in the area. In general, the suggested immunizing effect of LIT remains of immense interest considering both the high recurrence rates of BCC and SCC, as well as likelihood of new KC development.

Despite promising results, LIT remains an experimental strategy, and further research in the field is warranted. As BCC and SCC display distinct characteristics that might result in differences in their response to LIT, studies focusing on BCC and SCC separately are necessary. Currently, evidence on laser monotherapy is derived not only from studies on KC and KC models, but also other malignancies (including MM), non-KC tumor models and from healthy/photodamaged skin. Study findings may not be directly applicable to KC, but provide insights into the immunological responses following laser-tissue interaction and thus, enhance our basic understanding of LIT. Such studies furthermore help corroborate and extend what has been reported for KC.

Indicating that LIT is an evolving scientific field, 10 records currently appear on clinicaltrials.gov on cancer and laser immunotherapy, with five studies presently recruiting. Taking LIT from an experimental to a clinical setting, both SCC and BCC appear to be suitable indications of focus, since these skin tumors are typically small, local and easily accessible. Regardless of the often nonaggressive nature of KC, the morbidity of these skin cancers should not be underestimated. With a continuously increasing incidence combined with the fact that many patients develop multiple tumors over years, the demand for non-scarring treatment modalities is high. Specific clinical application of LIT could be as adjuvant to the treatment of KC patients that are non-responders to immunotherapy. Future studies should focus on which laser device is most optimal for LIT, as comparative studies of different laser modalities are not reported. Furthermore, local/intratumoral application of current exclusively systemic immunotherapy could be of great interest, in order to obtain high efficacy with mitigated side effects. Moreover, studies on KC and LIT aiming to prove sign of adaptive immunity in KC as seen in other cancer types are needed. Finally, optimal LIT treatment intervals remain to be elucidated, but may be expected to resemble current clinical immune check point inhibition treatment. The burgeoning field of LIT thus offers multiple avenues of investigation and conceivably, hitherto unlocked therapeutic potential.

## Figures and Tables

**Figure 1 cancers-13-05405-f001:**
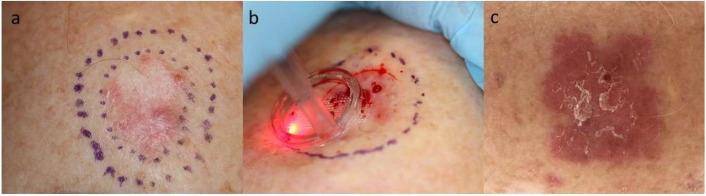
Photos of a BCC prior to AFL (**a**); upon intervention with AFL (**b**) and 1 week after AFL (**c**). The photo in the middle (**b**) shows the immediately generated laser channels (white grid in the area of the laser beam) on the skin. The latter photo (**c**) shows erythema of the AFL-treated area with resulting impact on the skin. Ultrapulse CO_2_-laser (10,600 nm, Lumenis, Santa Clara, CA, USA), 100 mJ/mb, 5% density). Department of Dermatology, Copenhagen University Hospital, Bispebjerg. Photos shown with patient’s consent.

**Figure 2 cancers-13-05405-f002:**
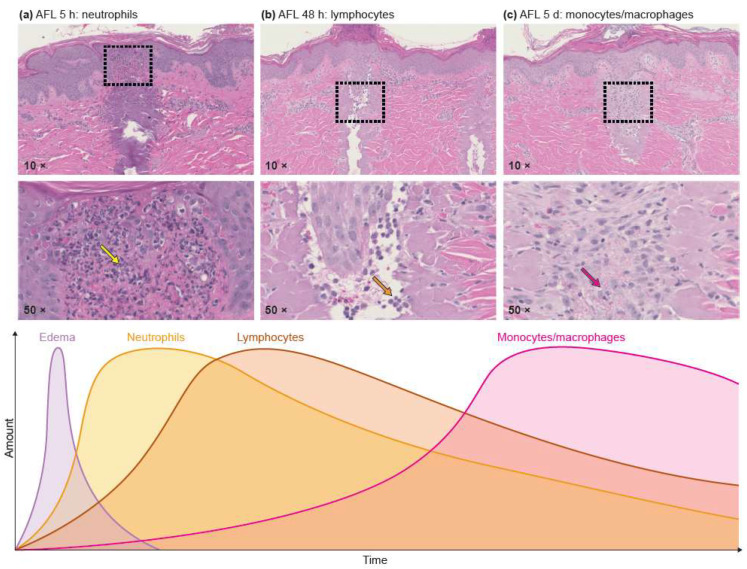
Histology of pigskin following AFL after respectively 5 h, 48 h and 5 days. The upper 3 photos (10× magnitude) illustrate the laser channels with influx of (**a**) neutrophilic granulocytes (**b**) lymphocytes and (**c**) macrophages. The lower three photos are the same as the upper but with magnitude 50× depicting the details of the attracted immune cells. The lower graph illustrates the timeframe for recruitment of the different immune cells where the colored peaks correspond to the colored arrows with influx of the specific immune cells in the IHC photos. Ultrapulse CO_2_ laser (10,600 nm, Lumenis, Santa Clara, CA, USA), 50 mJ/mb, 5% density). Wellman Center for Photomedicine, Massachusetts General Hospital, Boston, MA, USA.

**Figure 3 cancers-13-05405-f003:**
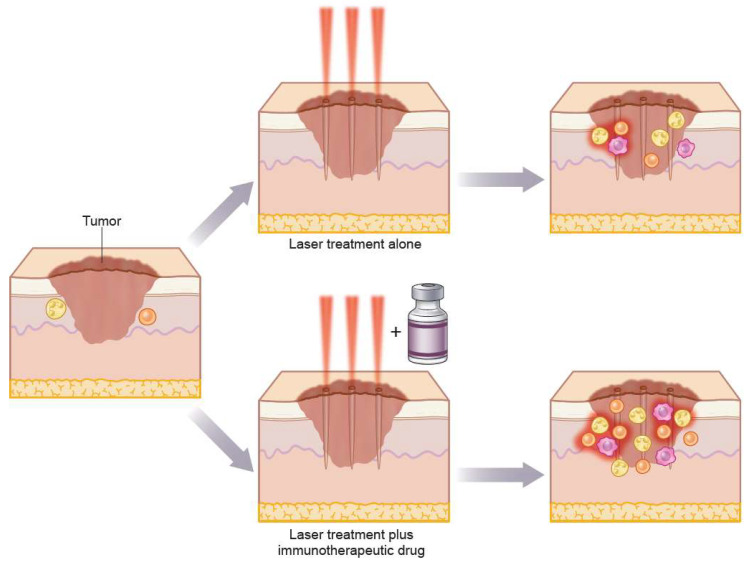
Conceptual summary illustrating the therapeutic benefits of laser monotherapy and LIT. The graphic illustration shows a keratinocyte carcinoma prior to laser intervention (left) and following laser monotherapy (upper right) and laser immunotherapy (LIT) (lower right). A moderate increase in immune cells following laser monotherapy is illustrated and the increased immune response of LIT compared with laser monotherapy is illustrated as a strong increase of immune cells following LIT. Yellow cells: neutrophilic granulocytes, orange cells: lymphocytes, purple cells: macrophages.

**Table 1 cancers-13-05405-t001:** Literature on laser monotherapy and immune activation.

Reference	Study Type	Laser Intervention	Major Findings
Kawakubo M et al. [34] 2017	Preclinical in vivo mouse tumor study	Fractional CO_2_ laser 100 mJ/mb 5% density	24 h Neutrophil infiltration 5 d: Abrogated increase of T-regs Tumor-specific CD8^+^ T-cell response (IHC & flow cytometry)
Wenande E et al. [35] 2018	Preclinical in vivo drug delivery pig study Extracted data on laser monotherapy from a larger study	Fractional CO_2_ laser 50 mJ/mb 5% density	48 h: neutrophil infiltration and perivascularlymphocytes 120 h: macrophage infiltration (H&E)
Fontenete S et al. [36] 2021	Preclinical in vivo mouse SCC-study Extracted data on laser monotherapy from a larger study	Fractional CO_2_ laser 100 mJ/mb 5% density	Tumor reduction Increased T-cell infiltration (IHC & flow cytometry)
DeBruler DM et al. [37] 2017	Preclinical in vivo porcine scar model	Fractional CO_2_ laser 70 mJ 5% density	1 h: increase of IL6 and MCP-1 Up to 168 h: increase of TGF-β (qRT-PCR)
Kawakubo M et al. [38] 2017	Preclinical in vivo mouse tumor study Extracted data on laser monotherapy from a larger study	Fractional CO_2_ laser 100 mJ/mb 5% density	Direct and indirect tumor response 5 d: increased T-cell infiltration and CD8/T-reg ratio (Flow cytometry)
Isbert C et al. [39] 2004	Preclinical in vivo rat tumor study comparing laser and surgery	Nd:Yag 1064 nm 2 W	Indirect tumor response of untreated tumor in laser but not surgery group. T-cell infiltration at untreated tumor border in laser group (IHC)
Helbig D et al. [40] 2009	Clinical tissue remodeling study on healthy individuals with photodamaged skin	Fractional CO_2_-laser 50, 64, 300 mJ 150 ablation zones per cm^2^	Day 3–14: increase of mast cells 1 h–3 days: increased TGFβ expression Day 3–14: increased HSPs, CD3, CD20, CD68 expression (IHC)
Prignano F et al. [41] 2009	Clinical tissue remodeling study on individuals with photodamaged skin	Fractional CO_2_-laser 2.07, 2.77, 4.15 J/cm^2^	Increased cytokine and growth factor infiltration with peak at day 3 with medium energy
Grunewald S et al. [42] 2011	Clinical study on individuals with photodamaged skin	Fractional CO_2_-laser 50 mJ, 100 mJ, 300 mJ	Increased lymphocyte infiltration from day 1 to 21 (H&E)
Odo LM et al. [43] 2011	Clinical study on individuals with normal skin	Fractional Er:YAG 1400 mJ/cm^2^	Progressive reduction of Langerhans cells, TLR 2&9 up to day 14 Increase of TLR3
Schmitt L et al. [44] 2017	Human 3D organotypic skin model Morphological and molecular changes upon laser	Fractional Er:YAG 4–10 J/cm2	48–72 h: upregulated expression of chemokines IL6,8,24 Downregulated expression of IL18,36β (qRT-PCR)

Abbreviations: Ablative fractional laser (AFL), Immunohistochemistry (IHC), Hematoxylin and eosin (H&E), Squamous cell carcinoma (SCC), Laser immunotherapy (LIT), Regulatory T-cells (T-regs), Monocyte chemoattractant protein-1 (MCP-1), Heat shock protein (HSP), Transforming growth factor-β (TGF-β), Toll like receptor (TLR), Er:YAG: Erbium YAG laser.

**Table 2 cancers-13-05405-t002:** Literature on laser immunotherapy (LIT) and immune activation.

Reference	Study Type	Laser Intervention and Immunostimulant	Major Findings
Chen W et al. [58] 1997	Preclinical in vivo rat tumor study	Laser: 805 nm diode 2–5 W Photosensitizer: Indocyanine green (ICG) Immunostimulant: Glycated chitosan gel (GC) (intratumoral injection)	Increase in survival rate, tumor eradication (primary and metastatic) Resistance to tumor rechallenge in successfully treated rats
Chen W et al. [59] 2001	Preclinical in vivo rat tumor study	Laser: 805 nm diode 2 W Photosensitizer: ICG Immunostimulant: GC (intratumoral injection)	LIT-cured rats showed total resistance to tumor reinoculation Spleen cells from LIT-cured rats provided 100% protection to the naïve recipient rats
Chen W et al. [60] 2002	Preclinical in vivo rat tumor study	Laser: 805 nm diode 2 W Photosensitizer: ICG Immunostimulant: GC (intratumoral injection)	Increased survival Reduced tumor-growth 3-component treatment was superior
Chen W et al. [61] 2000	Preclinical in vivo rat tumor study	Laser: 805 nm diode 2 W 1200 J Photosensitizer: ICG Immunostimulant: GC (intratumoral injection)	Lymphocyte and plasma cell infiltration (Electron microscopy, optical microscopy)
Luo M et al. [62] 2018	Preclinical in vivo mouse SCC study Clinical case on one patient with refractory SCC	Laser: 808-nm diode 1 W/cm^2^ Immunostimulant: Imiquimod (topical)	LIT-treated tumors showed no growth Increased survival Increased infiltration of lymphocytes in patient SCC and tumor reduction (H&E)
Kawakubo M et al. [38] 2017	Preclinical in vivo mouse tumor study	Laser: Fractional CO_2_ laser 100 mJ/mb 5% density Immunostimulant: Anti-PD1 (intraperitoneal injection)	Direct and indirect tumor response Systemic immunity observed Increased T-cell infiltration (Flow cytometry)
Fontenete S et al. [36] 2021	Preclinical in vivo mouse SCC-study	Laser: Fractional CO_2_ 100 mJ/mb 5% density Immunostimulant: Imiquimod (topical)	Direct tumor response AFL+imiquimod superior to monotherapy on tumor shrinkage and innate and adaptive immune cell recruitment (Flow cytometry & IHC)
Lo JA et al. [63] 2021	Preclinical in vivo mouse tumor study	Laser: Fractional CO_2_ 100 mJ/mb 5% density Immunostimulants: Anti-PD1 (systemic) Anti-CTLA4 (systemic) Imiquimod (topical)	Direct and indirect tumor response Superior efficacy with combination of imiquimod + AFL + anti–PD1 Increased CD8:Treg ratio (Flow cytometry)
Luo L et al. [64] 2018	Nanoparticle LIT preclinical in vivo mouse tumor study	Laser: near-infrared (NIR) 2 W/cm^2^ Immunostimulant: Anti-PD1 in nanoparticles (anti-PD1-NP) (intratumoral injection)	(anti-PD1-NP) + laser: stronger inhibition of tumor growth compared with (anti-PD1-NP) without laser Infiltration of T-cells (Flow cytometry)
Chen Q et al. [65] 2016	Nanoparticle LIT preclinical in vivo mouse tumor study Multiple groups with different interventions	Laser: NIR Immunostimulant: ICG + TLR (subcutaneous & IV) Anti-CTLA4 (IV injection) 0,5 W/cm^2^	Combinatory treatment led to increased DC and interleukin infiltration/release Almost completely inhibited growth of secondary tumor with combined laser-nano (ICG + TLR) and injected CTLA4 (Flow cytometry & ELISA)
Cao Q et al. [66] 2020	Nanoparticle preclinical in vivo mouse tumor study Multiple groups with different interventions	Laser: 1064 Q-switched Nd:Yag Copper monosulfide nanoparticles (Cus NP) 2,2 W/cm^2^ Immunostimulant: Anti-PD1 (IV and intraperitoneal injection) TLR (intratumoral injection)	Direct and indirect tumor response Increase in survival rate Infiltration of T-cells and dendritic cells (Flow cytometry)
Li X et al. [67] 2010	Clinical case study on 11 patients with late-stage MM	Laser: NIR 805 nm 1 W/cm^2^ Immunostimulant: Imiquimod (topical)	CR in 6/11 patients Complete local response in 8/11 patients AE: Rash, pruritus (H&E)
St Pierre et al. [68] 2010	Clinical case study on 64-year-old man with recurrence of MM on the toe	Laser: 810-nm diode 1 j/cm^2^ Immunostimulant: Imiquimod (topical)	Initial CR on treated tumors and PR of non-treated tumor but recurrence Repeated treatment episodes with continued response. No visceral metastases

Abbreviations: Laser immunotherapy (LIT), Indocyanine green (ICG), Glycated chitosan gel (GC), Squamous cell carcinoma (SCC), Regulatory T-cell (T-reg), Programmed death 1 (PD1), cytotoxic T-lymphocyte antigen 4 (CTLA4), Toll like receptor (TLR), Near infrared laser (NIR), IV: intravenous, Copper monosulfide nanoparticles (Cus-NP), Haematoxylin and eosin (H&E), Immunohistochemistry (IHC), Malignant melanoma (MM), Complete response (CR), Partial response (PR), Adverse event (AE).

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
