# Peer review of "Laser Immunotherapy: A Potential Treatment Modality for Keratinocyte Carcinoma"

_cancers, 2021, doi:10.3390/cancers13215405_

Round 1

Reviewer 1 Report

I would like to congratulate the authors on having done a great job with the revised version. I only have a couple of minor points for the authors to consider:

Line 99, adjust the font

Line 342-343, repetitive sentence consider revising

This manuscript is a resubmission of an earlier submission. The following is a list of the peer review reports and author responses from that submission.

Round 1

Reviewer 1 Report

This is an improved version of the review, with a thoughtful re-interpretation of the two tables and some background information on the laser effects. There remain, however, some confusing statements, and a conceptual summary is missing.

Abstract or Intro: As I assume this review is directed at a broader readership, it would be beneficial to define the relationship between KC, BSS, SCC  early on. Also, as it becomes clear in the subsequent sections, BCC and SCC respond quite differently to therapeutic interventions described in this review and perhaps should be regarded as separate entities for this purpose

Line 117 “A large case series presenting the genomic correlates on advanced/metastatic BCC treated with anti-PD1 revealed biological features (high TMB; PD-L1 amplification) predict beneficial potential of immunotherapy [16]”.  This study included only 8 patients with metastatic BCC, four of whom received anti-PD1- please include these vital statistics

Line 146, “in addition to cemiplimab studies, evidence of a clinically relevant potential for anti-PD1 treatment against KC has been reported in patients with MM on anti-PD1 treatment. In that population, lower incidences of BCC compared with patients with MM not receiving anti-PD1 was shown [21]”.  This study however, found no difference in SCC incidence – how does it fit with the presumed BCC v SCC biology? All other evidence quoted thus far is about locally advanced or metastatic BCC – would this justify separating KCs into at least two groups with a potentially different treatment approach when it comes to immunotherapy, such as immune checkpoint blockade?

Table 1. Please add interpretation where relevant (e.g. increase/decrease), e.g. Wenande et al,  major finding: “120 hrs: Macrophages”, and the site for in vivo models (e.g. DeBruler “increase in IL6” – local and/or systemic?)

Line 208, “As an acute immune infiltrator, IL6 signaling is important for activation, proliferation and survival of lymphocytes”  - do the authors refer to tumor-infiltrating lymphocytes or the cytokine? The protumorigenic role of IL-6 is well recognised; would the authors find some more recent/original references to support an immunogenic role for IL-6 in cancer immunity?  

Line 215, “altered T-reg ratios,” requires a reference population

Line 216, the relevance of colon carcinoma transplantable model is unclear

Line 248, Ref 35 – this is an interesting and one of the more relevant preclinical studies quoted in this review, thus more detailed description is warranted in my opinion (Table 2 provides still a very limited information about this study)

Line 298, “A body of more than 25  publications covering preclinical and clinical studies show substantial laser-mediated induction of specific antibodies…” – it would be great to provide some of these references, with a succinct summary of the critical findings

Line 308, “More conflicting immune reactions were noted in the clinical studies, all of which, were conducted in non-tumor tissue.” – would the authors please explain what they mean by this, and provide some references if relevant?

Ref 66 – although I understand the temptation of including melanoma studies, such studies do not come under the KC umbrella and perhaps would best be excluded from the review?

Finally, a figure providing a conceptual summary of the therapeutic benefits afforded by [laser + immunotherapy] combination, would be helpful

Minor points

Lines 95-98 (PD1/PD-L1 expression and high TMB for KC) - please reference all three statements

Figure 1 legend, Line 79, “Photos shown with patient’s accept.” Please include the consent statement/forms, as appropriate, at the final section, or a reference to the original publication, as appropriate

Fix reference 37

Reviewer 2 Report

The authors significantly improved the manuscript.